# Translation Synchronization via Truncated Least Squares

**Xiangru Huang**[*]
The University of Texas at Austin
2317 Speedway, Austin, 78712
xrhuang@cs.utexas.edu

**Zhenxiao Liang**[*]
Tsinghua University
Beijing, China, 100084
liangzx14@mails.tsinghua.edu.cn

**Chandrajit Bajaj**
The University of Texas at Austin
2317 Speedway, Austin, 78712
bajaj@cs.utexas.edu

**Qixing Huang**
The University of Texas at Austin
2317 Speedway, Austin, 78712
huangqx@cs.utexas.edu

## Abstract

In this paper, we introduce a robust algorithm, *TranSync*, for the 1D translation synchronization problem, in which the aim is to recover the global coordinates of a set of nodes from noisy measurements of relative coordinates along an observation graph. The basic idea of TranSync is to apply truncated least squares, where the solution at each step is used to gradually prune out noisy measurements. We analyze TranSync under both deterministic and randomized noisy models, demonstrating its robustness and stability. Experimental results on synthetic and real datasets show that TranSync is superior to state-of-the-art convex formulations in terms of both efficiency and accuracy.

## 1 Introduction

In this paper, we are interested in solving the 1D translation synchronization problem, where the input is encoded as an observation graph $\mathcal{G} = (\mathcal{V}, \mathcal{E})$ with $n$ nodes (i.e. $\mathcal{V} = \{1, \cdots, n\}$). Each node is associated with a latent coordinate $x_i^\star \in \mathbb{R}, 1 \leq i \leq n$, and each edge $(i, j) \in \mathcal{E}$ is associated with a noisy measurement $t_{ij} = x_i^\star - x_j^\star + \mathcal{N}(\epsilon_{ij})$ of the coordinate difference $x_i - x_j$ under some noise model $\mathcal{N}(\epsilon_{ij})$. The goal of translation synchronization is to recover the latent coordinates (up to a global shift) from these noisy pairwise measurements. Translation synchronization is a fundamental problem that arises in many application domains, including joint alignment of point clouds [7] and ranking from relative comparisons [8, 16].

A standard approach to translation synchronization is to solve the following linear program:

$$\text{minimize} \sum_{(i,j) \in \mathcal{E}} |t_{ij} - (x_i - x_j)|, \quad \text{subject to} \quad \sum_{i=1}^n x_i = 0, \tag{1}$$

Where the constraint ensures that the solution is unique. The major drawback of the linear programming formulation is that it can only tolerate up to 50% of measurements coming from biased noise models (e.g., uniform samples with non-zero mean). Moreover, it is challenging to solve (1) efficiently at scale. Solving (1) using interior point method becomes impractical for large-scale datasets, while more scalable methods such as coordinate descent usually exhibit slow convergence.

In this paper, we introduce a robust and scalable algorithm, TranSync, for translation synchronization. The algorithm is rather simple, we solve a truncated least squares at each iteration $k$:

$$\{x_i^{(k)}\} = \underset{\{x_i\}}{\operatorname{argmin}} \sum_{(i,j)\in\mathcal{E}} w_{ij}|t_{ij} - (x_i - x_j)|^2, \quad \text{subject to} \quad \sum_{i=1}^{n}\sqrt{d_i}x_i = 0, \quad d_i := \sum_{j\in\mathcal{N}(i)} w_{ij}.$$

(2)

where the weights $w_{ij} = Id(|t_{ij} - (x_i^{(k-1)} - x_j^{(k-1)})| < \delta_k)$ are obtained from the solution at the previous iteration using a geometrically decaying truncation parameter $\delta_k$. Although TranSync requires solving a linear system at each step, these linear systems are fairly similar to each other, meaning that the solution at the previous iteration provides an excellent warm-start for solving the linear system at the current iteration. As a result, the computational efficiency of TranSync is superior to state-of-the-art methods for solving the linear programming formulation. We analyze TranSync under both deterministic and randomized noise models, demonstrating its robustness and stability. In particular, we show that TranSync is able to handle biased noisy measurements.

We have evaluated TranSync on both synthetic datasets and real datasets used in the applications of joint alignment of point clouds and ranking from pair-wise measurements. Experimental results show that TranSync is superior to state-of-the-art solvers for the linear programming formulation in terms of both computational efficiency and accuracy.

## 1.1 Related Work

Translation synchronization falls into the general problem of map synchronization, which takes maps computed between pairs of objects as input, and outputs consistent maps across all the objects. Map synchronization appears as a crucial step in many scientific problems, including fusing partially overlapping range scans [15], assembling fractured surfaces [14], solving jigsaw puzzles [5, 11], multi-view structure from motion [25], data-driven shape analysis and processing [17], and structure from motion [27].

Early methods for map synchronization focused on applying greedy algorithms [14, 15, 18] or combinatorial optimization [20, 23, 27]. Although these methods exhibit certain empirical success, they lack theoretical understanding (e.g. we do not know under what conditions can the underlying ground-truth maps be exactly recovered).

Recent methods for map synchronization apply modern optimization techniques such as convex optimization and non-convex optimization. In [13], Huang and Guibas introduce a semidefinite programming formulation for permutation synchronization and its variants. Chen et al. [4] generalize the method to partial maps. In [26], Wang and Singer introduce a method for rotation synchronization. Although these methods provide tight, exact recovery conditions, the computational cost of the convex optimizations provide an obstruction for applying these methods to large-scale data sets.

In contrast to convex optimization, very recent map synchronization methods leverage non-convex optimization approaches such as spectral techniques and gradient-based optimization. In [21, 22], Pachauri et al. study map synchronization from the perspective of spectral decomposition. Recently, Shen et al. [24] provide an analysis of spectral techniques for permutation synchronization. Beyond spectral techniques, Zhou et al. [28] apply alternating minimization for permutation synchronization. Finally, Chen and Candes [3] introduce a method for the generalized permutation synchronization problem using the projected power method. To the best of our knowledge, we are the first to develop and analyze continuous map synchronizations (e.g., translations or rotations) beyond convex optimization.

Our approach can be considered as a special case of reweighted least squares (or RLS) [9, 12], which is a powerful method for solving convex and non-convex optimizations. The general RLS framework has been applied for map synchronization (e.g. see [1, 2]). Despite the empirical success of these approaches, the theoretical understanding of RLS remains rather limited. The analysis in this paper provides a first step towards the understanding of RLS for map synchronization.

## 1.2 Notation

Before proceeding to the technical part of this paper, we introduce some notation that will be used later. The unnormalized graph Laplacian of a graph $G$ is denoted as $L_G$. If it is obvious from the

---

**Algorithm 1** TranSync($c, k_{\max}$)

---

1. $\mathbf{x}^{(-1)} \leftarrow \mathbf{0}$. $\delta_{-1} \leftarrow \infty$.
**for** $k = 0, 1, 2, k_{\max}$ **do**
   2. Obtain the truncated graph $\mathcal{G}^{(k)}$ using $\mathbf{x}^{(k-1)}$ and $\delta_{k-1}$.
   3. **Break if** $\mathcal{G}^{(k)}$ is disconnected
   4. Solve (2) using (4) to obtain $\mathbf{x}^{(k)}$.
   5. $\delta_k = \min \big( \max_{(i,j) \in \mathcal{E}} |t_{ij} - (x_i^{(0)} - x_j^{(0)})|, c\delta_{k-1} \big)$.
**end for**
**Output:** $\mathbf{x}^{(k)}$.

---

context, we will always shorten $L_G$ as $L$ to make the notation uncluttered. Similarly, we will use $D = \mathrm{diag}(d_1, \cdots, d_n)$ to collect the vertex degrees and denote the vertex adjacency and vertex-edge adjacency matrices as $A$ and $B$ respectively. The peusdo-inverse of a matrix $X$ is given by $X^+$. In addition, we always sort the eigenvalues of a symmetric matrix $X \in \mathbb{R}^{n \times n}$ in increasing order (i.e. $\lambda_1(X) \le \lambda_2(X) \le \cdots \le \lambda_n(X)$). Moreover, we will consider several matrix norms $\|\cdot\|, \|\cdot\|_{1,\infty}$ and $\|\cdot\|_{\mathcal{F}}$, which are defined as follows:

$$\|X\| = \sigma_{\max}(X), \quad \|X\|_{1,\infty} = \max_{1 \le i \le n} \sum_{j=1}^{n} |x_{ij}|, \quad \|X\|_{\mathcal{F}} = \Big( \sum_{i,j} x_{ij}^2 \Big)^{\frac{1}{2}}.$$

Note that $\|X\|_{1,\infty}$ is consistent with the $L^\infty$-norm of vectors.

## 2 Algorithm

In this section, we provide the algorithmic details of TranSync. The iterative scheme (1) requires an initial solution $\mathbf{x}^{(0)}$, an initial truncation parameter $\delta_0$, and a stopping condition. The initial solution can be determined by solving for $\mathbf{x}^{(0)}$ from (2) w.r.t. $w_{ij} = 1$. We set the initial truncation parameter $\delta_0 = \max_{(i,j) \in \mathcal{E}} |t_{ij} - (x_i^{(0)} - x_j^{(0)})|$, so that the edge with the biggest residual is removed. We stop TranSync either after the maximum number of iterations is reached, or the truncated graph becomes disconnected. Algorithm 1 provides the pseudo code of TranSync.

Clearly, the performance of TranSync is driven by the efficiency of solving (2) at each iteration. TranSync takes an iterative approach, in which we utilize a warm-start $\mathbf{x}^{(k-1)}$ provided by the solution obtained at the previous iteration. When the truncated graph is non-bipartite, we find a simple weighted average scheme delivers satisfactory computational efficiency. Specifically, it generates a series of vectors $\mathbf{x}^{k,0} = \mathbf{x}^{(k-1)}, \mathbf{x}^{k,1}, \cdots, \mathbf{x}^{k,n_{\max}}$ via the following recursion:

$$\overline{x}_i^{k,l+1} = (1 - \epsilon) \sum_{j \in \mathcal{N}(i)} w_{ij}(x_j^{k,l} + t_{ij}) / \sum_{j \in \mathcal{N}(i)} w_{ij} + \epsilon \overline{x}_i^{k,l} \tag{3}$$

$$x_i^{k,l+1} = \overline{x}_i^{k,l+1} - \frac{1}{\sum_{i'=1}^{n} \sqrt{d_i}} \sum_{i'=1}^{n} \sqrt{d_{i'}} \, \overline{x}_{i'}^{k,l+1}, \tag{4}$$

which may be written in the following matrix form:

$$\mathbf{x}^{k,l+1} = (I_n - \frac{1}{n} D^{-\frac{1}{2}} \mathbf{1} \mathbf{1}^T D^{\frac{1}{2}})[(1 - \epsilon)D^{-1}\big(A\mathbf{x}^{k,l} + B\mathbf{t}^{(k)}\big) + \epsilon \mathbf{x}^{k,l}], \tag{5}$$

Here we add the parameter $\epsilon$ to create a small perturbation to avoid the special case of bipartite graphs. For non-bipartite graphs, $\epsilon$ can be set to zero.

**Remark 2.1** *The corresponding normalization constraint in (4), i.e., $\sum_i \sqrt{d_i}x_i = 0$, only changes the solution to (2) by a constant factor. We utilize this modification for the purpose of obtaining a concise convergence property of the iterative scheme detailed below.*

The following proposition states that (4) admits a geometric convergence rate:

**Proposition 2.1** $\boldsymbol{x}^{k,l}$ *geometrically converges to* $\boldsymbol{x}^{(k+1)}$*. Specifically,* $\forall l \geq 0$,

$$\|D^{\frac{1}{2}}\left(\boldsymbol{x}^{k,l} - \boldsymbol{x}^{(k)}_{\text{shift}}\right)\| \leq (1 - (1-\epsilon)\rho)^l \|D^{\frac{1}{2}}\left(\boldsymbol{x}^{k,0} - \boldsymbol{x}^{(k)}_{\text{shift}}\right)\|, \quad \boldsymbol{x}^{(k)}_{\text{shift}} = \boldsymbol{x}^{(k)} - \frac{\sum_i \sqrt{d_i}x_i^{(k)}}{\sum_i \sqrt{d_i}}\boldsymbol{1}.$$

*where* $\rho < 1$ *is the spectral gap of the normalized Graph Laplacian of the truncated graph.*

*Proof.* See Appendix A.

Since the intermediate solutions are mainly used to prune outlier observations, it is clear that $O(\log(n))$ iterations of (5), which induce a $O(1/n)$ error for solving (2), are sufficient. The complexity of checking if the graph is non-bapriatite is $O(|\mathcal{E}|)$. The total running time for solving (2) is thus $O(|\mathcal{E}|\log(n))$. This means the total running time of TranSync is $O(|\mathcal{E}|\log(n)k_{\max})$, making it scalable to large-scale datasets.

# 3 Analysis of TranSync

In this section, we provide exact recovery conditions of TranSync. We begin with describing an exact recovery condition under a deterministic noise model in Section 3.1. We then study an exact recovery condition to demonstrate that TranSync can handle biased noisy samples in Section 3.2.

## 3.1 Deterministic Exact Recovery Condition

We consider the following deterministic noisy model: We are given the ground-truth location $\mathbf{x}^{gt}$. Then, for each correct measurement $t_{ij}, (i,j) \in \mathcal{G}$, $|t_{ij} - (x_i^{gt} - x_j^{gt})| \leq \sigma$ for a threshold $\sigma$. In contrast, each incorrect measurement $t_{ij}, (i,j) \in \mathcal{G}$ could take any real number. The following theorem provides an exact recovery condition under this noisy model.

**Theorem 3.1** *Let* $d_{bad}$ *be the maximum number of incorrect measurements per node. Define*

$$\alpha = \max_k L^{\dagger}_{G,kk} + \max_{i \neq j} L^{\dagger}_{G,ij} + \frac{n}{2} \max_{\substack{i,j,k \\ \text{pairwisely different}}} |L^{\dagger}_{G,ki} - L^{\dagger}_{G,kj}|,$$

*and*

$$h = \alpha d_{bad}, \quad p = \frac{d_{bad}\alpha}{1 - 2h}, \quad q = \frac{(n - d_{bad})\alpha}{1 - 2h}.$$

*Suppose* $h < \frac{1}{6}$ *(or* $p < \frac{1}{4}$*), then starting from any initial solution* $\boldsymbol{x}^{(0)}$*, and for any large enough initial truncation threshold* $\epsilon \geq 2\|\boldsymbol{x}^{(0)}\|_{\infty} + \sigma$ *and iterative step size* $c$ *satisfying* $4p < c < 1$*, we have*

$$\|\boldsymbol{x}^{(k)} - \boldsymbol{x}^{gt}\|_{\infty} \leq q\sigma + 2p\epsilon c^{k-1},$$

*where*

$$k \leq -\log\left(\frac{\epsilon(c - 4p)}{(1 + 2q)\sigma}\right)/\log c + 1.$$

*Moreover, we can eventually reach an* $x^{(k)}$ *such that*

$$\|\boldsymbol{x}^{(k)}\|_{\infty} \leq \frac{2p + cq}{c - 4p}\sigma$$

*which is independent of the initial solution* $\boldsymbol{x}^{(0)}$*, initial truncation threshold* $\epsilon$*, and values of all wrong measurements* $\boldsymbol{t}_{G \setminus G_{good}}$*.*

*Proof:* See Appendix B. □

Theorem 3.1 essentially says that TransSync can tolerate a constant fraction of arbitrary noise. To understand how strong this condition is, we consider the case where $G = \mathcal{K}_n$ is given by a clique. Moreover, we assume the nodes are divided into two clusters of equal size, where all the measurements within each cluster are correct. For measurements between different clusters, half of them are correct and the other half are wrong. In this case, 25% of all measurements are wrong. However, we cannot recover the original $\mathbf{x}^{gt}$ in this case. In fact, we can set the wrong measurements in a consistent

manner, i.e $t_{ij} = x_i^{gt} - x_j^{gt} + b$ for a constant $b \neq 0$, leading to two competing clusters (one correct and the other one incorrect) with equal strength. Hence, in the worst case, any algorithm can only tolerate at most 25% of measurements being wrong.

We now try to use Theorem 3.1 to analyze the case where the observation graph is a clique. In this case, it is clear that $\alpha = \frac{1}{n}$, and $p = \frac{d_{bad}}{n}$, i.e the fraction of wrong measurements out of all measurements. Hence, in the clique case, we have shown that TranSync converges to a neighborhood of the ground truth from any initial solution if the fraction of wrong measurements is less that $\frac{1}{6}$ (i.e., 2/3 of the upper bound).

## 3.2 Biased Random Noisy Model

We proceed to provide an exact recovery condition of TranSync under a biased random noisy model. To simplify the discussion, we assume the observation graph $\mathcal{G} = \mathcal{K}_n$ is a clique. However, our analysis framework can be extended to handle arbitrary graphs.

Assume $\sigma << a + b$. We consider the following noise model, where the noisy measurements are independent, and they follow

$$
t_{ij} = \begin{cases} x_i^{gt} - x_j^{gt} + U[-\sigma, \sigma] & \text{with probability } p \\ x_i^{gt} - x_j^{gt} + U[-a, b] & \text{with probability } 1 - p \end{cases}
\tag{6}
$$

It is easy to check that the linear programming formulation is unable to recover the ground-truth solution if $\frac{b}{a+b}(1 - p) > \frac{1}{2}$. The following theorem shows that TranSync achieves a sub-constant recovery rate instead.

**Theorem 3.2** *There exists a constant c so that if $p > c/\sqrt{\log(n)}$, then w.h.p,*

$$
\|\boldsymbol{x}^{(k)} - \boldsymbol{x}^{gt}\|_\infty \leq (1 - p/2)^k (b - a), \qquad \forall \, k = 0, \cdots, [-\log(\frac{b + a}{2\sigma})/log(1 - p/2)].
$$

The major difficulty of proving Theorem 3.2 is that $\mathbf{x}^{(k)}$ is dependent on $\mathbf{t}_k$, making it hard to control $\mathbf{x}^{(k)}$ using existing concentration bounds. We address this issue by showing that the solutions $\mathbf{x}^{(k)}, k = 0, \cdots$, stay close to the segment between $x^{gt}$ and $x^{gt} + (1 - p)\frac{a+b}{2}\mathbf{1}$. Specifically, for points on this segment, we can leverage the independence of $t_{ij}$ to derive the following concentration bound for one step of TranSync:

**Lemma 3.1** *Consider a fixed observation graph $\mathcal{G}$. Let $\bar{r} = \frac{(a+b)p}{(a+b)p + 2(1-p)\delta}$ and $d_{\min}$ be the minimum degree of $\mathcal{G}$. Suppose $d_{\min} = \Omega(log^2(n))$, and $p + \bar{r}(1 - p) = \Omega(\log^2(n)/d_{\min})$ . Consider an initial point $\boldsymbol{x}^{(0)}$ (independent from $t_{ij}$) and a threshold parameter $\delta$ such that $-a + \delta \leq \min_i x_i^{(0)} \leq \max_i x_i^{(0)} \leq b - \delta$. Then w.h.p., one step of TranSync outputs $\boldsymbol{x}^{(1)}$ which satisfies*

$$
\|\boldsymbol{x}^{(1)} - ((1 - \bar{r})\boldsymbol{x}^{(0)} + \bar{r}\boldsymbol{x}^{gt})\|_\infty
$$
$$
= O\left(\sqrt{\frac{\log(n)}{(p + \bar{r}(1 - p))d_{\min}\lambda_2(\overline{L}_\mathcal{G})}}\right) \cdot \sqrt{\max(\|\boldsymbol{x}^{(0)}\|_{d,\infty}^2, \bar{r}^2) + O\left(\frac{p}{\bar{r}}\sigma^2\right)},
$$

*where $\|\boldsymbol{x}^{(0)}\|_{d,\infty} = \max_{1 \leq i,j \leq n} |x_i^{(0)} - x_j^{(0)}|$, and $\overline{L}_\mathcal{G}$ is the normalized graph Laplacian of $\mathcal{G}$.*

*Proof:* See Appendix C.1. □

**Remark 3.1** *Note that when $\mathcal{G}$ is a clique or a graph sampled from the standard Erdős-Rényi model $G(n, q)$, then $O(\sqrt{\frac{\rho \log(n)}{(p+\bar{r}(1-p))\lambda_2(L_\mathcal{G})}}) = O(\sqrt{\frac{\log(n)}{(p+\bar{r}(1-p))n}})$.*

To prove Theorem 3.2, we show that when $\bar{k} = O(\log^{\frac{3}{4}}(n))$, the $L^\infty$ distance between $\mathbf{x}^{(k)}$ to the line segment between $x^{gt}$ and $x^{gt} + (1 - p)\frac{a+b}{2}\mathbf{1}$ only grows geometrically, and this distance is in the order of $o(p)$. On the other hand, $(1 - p/2)^{\bar{k}} = o(p)$. So when $k \geq \bar{k}$, that distance decays with a geometrical rate that is small than $c$. The details are deferred to Appendix C.2.

**Improving recovery rate via sample splitting.** Note that Lemma 3.1 enables us to apply standard sampling tricks to improve the recovery rate. To simplify the discussion, we will assume $\sigma$ is sufficiently small. First of all, it is clear that if re-sampling is allowed at each iteration, then TranSync admits a recovery rate of $O(\frac{\log(n)}{\sqrt{d_{min}}})$. When re-sampling is not allowed, we can improve the recovery rate by dividing the observations into $O(\frac{\log(n)}{\sqrt{n}})$ independent sets, and apply one set of observations at each iteration. In this case, the recovery rate is $O(\frac{\log^2(n)}{\sqrt{n}})$. These recovery rates suggest that the recovery rate in Theorem 3.2 could potentially be improved. Nevertheless, Theorem 3.2 still shows that TranSync can tolerate a sub-constant recovery rate, which is superior to the linear programming formulation.

# 4 Experimental Results

In this section, we provide a detailed experimental evaluation of the proposed translation synchronization (TranSync) method. We begin with describing the experimental setup in Section 4.1. We then perform evaluations on synthetic and real datasets in Section 4.2 and Section 4.3 respectively.

## 4.1 Experimental Setup

**Datasets.** We employ both synthetic datasets and real datasets for evaluation. The synthetic data is generated following the noisy model described in (6). In the following, we encode the noisy model as $\mathcal{M}(\mathcal{G}, p, \sigma)$, where $\mathcal{G}$ is the observation graph, $p$ is the fraction of correct measurements, and $\sigma$ describes the interval of correct measurements. Besides the synthetic data, we also consider two real datasets coming from the applications of joint alignment of point clouds and global ranking from relative rankings.

**Baseline comparison.** We choose coordinate descent for solving (1) as the baseline algorithm. Specifically, denote the solution of $x_i, 1 \leq i \leq n$ at iteration $k$ as $x_i^{(k)}$. Then $\{x_i^{(k)}\}$ are given by the following recursion:

$$x_i^{(k)} = \underset{x_i}{\arg\min} \sum_{j \in \mathcal{N}(i)} |x_i - (x_j^{(k-1)} - t_{ij})|$$
$$= \underset{j \in \mathcal{N}(i)}{\text{median}} \{x_j^{(k-1)} - t_{ij}\}, \qquad 1 \leq i \leq n, \qquad k = 1, 2, \cdots, \tag{7}$$

We use the same initial starting point as TranSync. We also tested interior point methods, and all the datasets used in our experiments are beyond their reach.

**Evaluation protocol.** We report the min, median, and max of the coordinate-wise difference between the solution of each algorithm and the underlying ground-truth. We also report the total running time of each algorithm on each dataset (See Table 1).

## 4.2 Experimental Evaluation on Synthetic Datasets

We generate the synthetic datasets by sampling from four kinds of observation graphs and two values of $\sigma$, i.e. $\sigma \in \{0.01, 0.04\}$. The graphs are generated according to two modes: 1) *dense* graphs versus *sparse* graphs, and 2) *regular* graphs versus *irregular* graphs. To illustrate the strength of TranSync, we choose $p \in \{0.4, 0.8\}$ for dense graphs and $p \in \{0.8, 1.0\}$ for sparse graphs. Below is a detailed descriptions for all kinds of observation graphs generated.

- $G_{dr}$ (dense, regular): The first graph contains $n = 2000$ nodes. Independently, we connect an edge between a pair of vertices $v_i, v_j$ with a fixed probability $p = 0.1$. The expected degree of each vertex is 200.

- $G_{di}$ (dense, irregular): The second graph contains $n = 2000$ nodes. Independently, we connect an edge between a pair of vertices $v_i, v_j$ with probability $p = 0.4 s_i s_j$, where $s_i = 0.2 + 0.6 \frac{i-1}{n-1}, 1 \leq i \leq n$ are scalar values associated the vertices. The expected degree of each vertex is about 200.

| $\mathcal{G}$ | $p$ | $\sigma$ | Coordinate Descent | | | | TranSync | | | |
|---|---|---|---|---|---|---|---|---|---|---|
| | | | min | median | max | time | min | median | max | time |
| $G_{dr}$ | 0.4 | 0.01 | 0.95e-2 | 1.28e-2 | 11.40e-2 | 0.939s | 0.30e-2 | 0.37e-2 | 0.60e-2 | 0.178s |
| $G_{dr}$ | 0.4 | 0.04 | 3.87e-2 | 4.73e-2 | 18.59e-2 | 1.325s | 1.04e-2 | 1.22e-2 | 1.59e-2 | 0.155s |
| $G_{dr}$ | 0.8 | 0.01 | 0.30e-2 | 0.34e-2 | 0.41e-2 | 0.781s | 0.16e-2 | 0.18e-2 | 0.28e-2 | 0.149s |
| $G_{dr}$ | 0.8 | 0.04 | 1.19e-2 | 1.35e-2 | 1.78e-2 | 1.006s | 0.57e-2 | 0.70e-2 | 0.87e-2 | 0.133s |
| $G_{di}$ | 0.4 | 0.01 | 2.17e-2 | 17.59e-2 | 50.51e-2 | 0.865s | 0.39e-2 | 0.52e-2 | 0.93e-2 | 0.179s |
| $G_{di}$ | 0.4 | 0.04 | 5.46e-2 | 19.40e-2 | 53.88e-2 | 1.043s | 1.25e-2 | 1.55e-2 | 2.42e-2 | 0.169s |
| $G_{di}$ | 0.8 | 0.01 | 0.34e-2 | 0.42e-2 | 0.58e-2 | 0.766s | 0.17e-2 | 0.24e-2 | 0.33e-2 | 0.159s |
| $G_{di}$ | 0.8 | 0.04 | 1.39e-2 | 1.66e-2 | 2.30e-2 | 0.972s | 0.68e-2 | 0.86e-2 | 1.16e-2 | 0.141s |
| $G_{sr}$ | 0.8 | 0.01 | 0.58e-2 | 0.65e-2 | 0.79e-2 | 10.062s | 0.38e-2 | 0.45e-2 | 0.61e-2 | 1.852s |
| $G_{sr}$ | 0.8 | 0.04 | 2.35e-2 | 2.62e-2 | 3.54e-2 | 12.375s | 1.35e-2 | 1.55e-2 | 2.05e-2 | 1.577s |
| $G_{sr}$ | 1.0 | 0.01 | 0.45e-2 | 0.50e-2 | 0.58e-2 | 9.798s | 0.28e-2 | 0.32e-2 | 0.39e-2 | 0.188s |
| $G_{sr}$ | 1.0 | 0.04 | 1.84e-2 | 1.99e-2 | 2.36e-2 | 11.626s | 1.14e-2 | 1.29e-2 | 1.60e-2 | 0.179s |
| $G_{si}$ | 0.8 | 0.01 | 0.72e-2 | 0.85e-2 | 75.85e-2 | 10.236s | 0.52e-2 | 0.64e-2 | 1.10e-2 | 1.835s |
| $G_{si}$ | 0.8 | 0.04 | 2.88e-2 | 3.38e-2 | 11.48e-2 | 12.350s | 1.79e-2 | 2.16e-2 | 3.59e-2 | 1.610s |
| $G_{si}$ | 1.0 | 0.01 | 0.53e-2 | 0.62e-2 | 0.77e-2 | 9.388s | 0.37e-2 | 0.43e-2 | 0.57e-2 | 0.180s |
| $G_{si}$ | 1.0 | 0.04 | 2.24e-2 | 2.52e-2 | 3.12e-2 | 12.200s | 1.44e-2 | 1.72e-2 | 2.47e-2 | 0.187s |

Table 1: Experimental results comparing TranSync and Coordinate Descent (CD) under different settings. All statistics (min, median, max) and mean running time are computed among 100 independent experiments with the same setting. As observed, TranSync outperforms Coordinate Descent in all experiments.

- $G_{sr}$ (sparse, regular): The third graph is generated in a similar fashion as the first graph, except that the number of nodes $n = 20K$, and the connecting probability is set to $p = 0.003$. The expected degree of each vertex is 60.
- $G_{si}$ (sparse, irregular): The fourth graph is generated in a similar fashion as the second graph, except that the number of nodes $n = 20K$, and the connecting probability between a pair of vertices is $p = 0.1s_i s_j$, where $s_i = 0.07 + 0.21\frac{i-1}{n-1}, 1 \le i \le n$ are scalar values associated the vertices. The expected degree of each vertex is about 60.

For this experiment, instead of using $k_{\max}$ as stopping condition as in Algorithm 1, we stop when we observe $\delta_k < \delta_{\min}$. Here $\delta_{\min}$ does not need to be close to $\sigma$. In fact, we choose $\delta_{\min} = 0.05, 0.1$ for $\sigma = 0.01, 0.04$, respectively. We also claim that if a small validation set (with size significantly less than $n$) of correct observations is available, our performance could be further improved.

As illustrated in Table 1, TranSync dominates coordinate descent in terms of both accuracy and prediction. In particular, TranSync is significantly better than coordinate descent on dense graphs in terms of accuracy. In particular, on dense but irregular graphs, coordinate descent did not converge at all when $p = 0.8$. The main advantage of TranSync on sparse graphs is the computational cost, although the accuracy is still considerably better than coordinate descent.

### 4.3 Experimental Evaluation on Real Datasets

**Translation synchronization for joint alignment of point clouds.** In the first application, we consider the problem of joint alignment of point clouds from pair-wise alignment [10]. To this end, we utilize the Patriot Circle Lidar dataset[1]. We uniformly subsampled the dataset to 6K scans. We applied Super4PCS [19] to match each scan to 300 randomly selected scans, where each match returns a pair-wise rigid transformation and a score. We then pick the top-30 matches for each scan, this results in a graph with 140K edges. To create the input data for translation synchronization, we run the state-of-the-art rotation synchronization algorithm described in [2] to estimate a global pose $R_i$ for each scan. The pair-wise measurement $\mathbf{t}_{ij}$ from node $i$ to node $j$ is then given by $R_i^T \mathbf{t}_{ij}^{\mathrm{local}}$, where $\mathbf{t}_{ij}^{\mathrm{local}}$ is the translation vector obtained in pair-wise matching. The average outlier ratio of the pair-wise matches per node is 35%, which is relatively high since the observation graph is fairly sparse. Since $\mathbf{t}_{ij}$ is a 3D vector, we run TranSync three times, one for each coordinate. As illustrated in Figure 1, TranSync is able to recover the the global shape of the underlying scanning trajectory. In contrast, coordinate descent completely fails on this dataset.

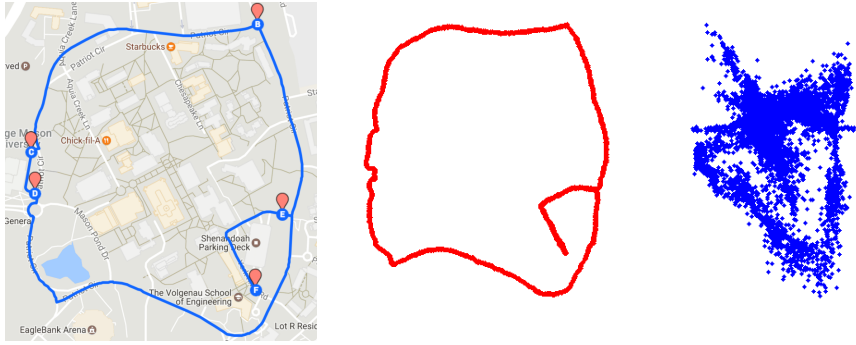

Figure 1: The application of TranSync in joint alignment of 6K Lidar scans around a city block. (a) Snapshot of the underlying scanning trajectory. (b) Reconstruction using TranSync (c) Reconstruction using Coordinate Descent.

| Movie | Global ranking (score) | | | | | |
|---|---|---|---|---|---|---|
| | MRQE | Hodge-Diff. | Hodge-Ratio | Hodge-Binary | TS-Init | TS-Final |
| Shakespeare in Love | 1(85) | 1(0.247) | 2(0.078) | 1 (0.138) | 1(0.135) | 1(0.219) |
| Witness | 2(77) | 2(0.217) | 1(0.088) | 3(0.107) | 3(0.076) | 2(0.095) |
| October Sky | 3(76) | 3(0.213) | 3(0.078) | 2(0.111) | 2(0.092) | 3(0.0714) |
| The Waterboy | 4(66) | 6(-0.464) | 6(-0.162) | 6(-0.252) | 5(-0.134) | 4(-0.112) |
| Interview with the Vampire | 5(65) | 4(-0.031) | 4(-0.012) | 4(-0.120) | 4 (-0.098) | 5(-0.140) |
| Dune | 6(44) | 5(-0.183) | 5(-0.069) | 5(-0.092) | 6(-0.216) | 6(-0.281) |

Table 2: Global ranking of selected six movies via different methods: MRQE, HodgeRank[16] with 1) arithmetic mean score difference, 2) geometric mean score ratio and 3) and binary comparisons, and the initial and final predictions of TranSync. TranSync results in the most consistent result with MRQE.

**Ranking from relative comparisons.** In the second application, we apply TranSync to predict global rankings of Netflix movies from their relative comparisons provided by users. The Netflix dataset contains 17070 movies that were rated between October, 1998 and December, 2005. We adapt the procedure described in [16] to generate the input data. Specifically, for each pair of movies, we average the relative ratings from the same users within the same month. We only consider a relative measurement if we collect more than 10 such relative ratings. We then apply TranSync to predict the global rankings of all the movies. We report the initial prediction obtained by the first step of TranSync (i.e., all the relative comparisons are used) and the final prediction suggested by TranSync (i.e., after removing inconsistent relative comparisons).

Table 2 compares TranSync with HodgeRank [16] on six representative movies that are studied in [16]. The experimental results show that both predictions appear to be more consistent with MRQE[2] (the largest online directory of movie reviews on the internet) than HodgeRank [16] and its variants, which were only applied on these six movies in isolation. Moreover, the final prediction is superior to the initial prediction. These observations indicate two key advantages of TranSync, i.e., scalability on large-scale datasets and robustness to noisy relative comparisons.

## 5 Conclusions and Future Work

In this paper, we have introduced an iterative algorithm for solving the translation synchronization problem, which estimates the global locations of objects from noisy measurements of relative locations. We have justified the performance of our approach both experimentally and theoretically under both deterministic and randomized conditions. Our approach is more scalable and accurate than the standard linear programming formulation. In particular, when the pair-wise measurement

is biased, our approach can still achieve sub-constant recovery rate, while the linear programming approach can tolerate no more than 50% of the measurements being biased.

In the future, we plan to extend this iterative scheme to other synchronization problems, such as synchronizing rotations and point-based maps. Moreover, it would also be interesting to study variants of the iterative scheme such as re-weighted least squares. We would also like to close the gap between the current recovery rate and the lower bound, which exhibits a poly-log factor. This requires developing new tools for analyzing the iterative algorithm.

**Acknowledgement.** Qixing Huang would like to acknowledge support this research from NSF DMS-1700234. Chandrajit Bajaj would like to acknowledge support for this research from the National Institute of Health grants #R41 GM116300 and #R01 GM117594.

## Footnotes

[1] http://masc.cs.gmu.edu/wiki/MapGMU

[2]http://www.mrqe.com

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
