[Supplementary Material · NIPS_2017_appendix.pdf]

# A Proof of Proposition 2.1

Denote vector $\mathbf{v}$ as $(\sqrt{d_1}, \sqrt{d_2}, \ldots, \sqrt{d_n})^T$. To shift any vector $\mathbf{x}$ so that $\mathbf{v}^T\mathbf{x} = 0$, we only need to multiply $(I - \frac{\mathbf{1}\mathbf{v}^T}{\mathbf{v}^T\mathbf{1}})$ on the left. For any $k$, we have

$$\mathbf{x}^{(k)} = L^\dagger B \mathbf{t}^{(k)}, \quad \mathbf{x}^{(k)}_{\text{shift}} = (I - \frac{\mathbf{1}\mathbf{v}^T}{\mathbf{v}^T\mathbf{1}})\mathbf{x}^{(k)} \tag{8}$$

Based on the algorithm (see (4)), we also know

$$\begin{align}
\bar{\mathbf{x}}^{k,l+1} &= (1-\epsilon)D^{-1}(A\mathbf{x}^{k,l} + B\mathbf{t}^{(k)}) + \epsilon\mathbf{x}^{k,l} \tag{9}\\
&= (1-\epsilon)D^{-1}(D-L)\mathbf{x}^{k,l} + (1-\epsilon)D^{-1}B\mathbf{t}^{(k)} + \epsilon\mathbf{x}^{k,l} \quad (\ A = D - L\ ) \tag{10}\\
&= (1-\epsilon)(I - D^{-1}L)\mathbf{x}^{k,l} + (1-\epsilon)D^{-1}L\mathbf{x}^{(k)} + \epsilon\mathbf{x}^{k,l} \quad (\text{ from (8) }) \tag{11}\\
&= \mathbf{x}^{k,l} - (1-\epsilon)D^{-1}L\mathbf{x}^{k,l} + (1-\epsilon)D^{-1}L\mathbf{x}^{(k)} \tag{12}\\
&= \mathbf{x}^{k,l} - (1-\epsilon)D^{-1}L\mathbf{x}^{k,l} + (1-\epsilon)D^{-1}L\mathbf{x}^{(k)}_{\text{shift}} \quad (\text{since } \mathbf{L1 = 0}) \tag{13}\\
&= \mathbf{x}^{k,l} - (1-\epsilon)D^{-1}L(\mathbf{x}^{k,l} - \mathbf{x}^{(k)}_{\text{shift}}) \tag{14}
\end{align}$$

This immediately implies

$$\begin{align}
\mathbf{x}^{k,l+1} - \mathbf{x}^{(k)}_{\text{shift}} &= (I - \frac{\mathbf{1}\mathbf{v}^T}{\mathbf{v}^T\mathbf{1}})(\mathbf{x}^{k,l} - \mathbf{x}^{(k)}) \tag{15}\\
&= (I - \frac{\mathbf{1}\mathbf{v}^T}{\mathbf{v}^T\mathbf{1}})(\mathbf{x}^{k,l} - \mathbf{x}^{(k)}_{\text{shift}}) \tag{16}\\
&= (I - \frac{\mathbf{1}\mathbf{v}^T}{\mathbf{v}^T\mathbf{1}})(I - (1-\epsilon)D^{-1}L)(\mathbf{x}^{k,l} - \mathbf{x}^{(k)}_{\text{shift}}) \tag{17}
\end{align}$$

or more related to the proposition, we have

$$D^{\frac{1}{2}}(\mathbf{x}^{k,l+1} - \mathbf{x}^{(k)}_{\text{shift}}) = [D^{\frac{1}{2}}(I - \frac{\mathbf{1}\mathbf{v}^T}{\mathbf{v}^T\mathbf{1}})(I - (1-\epsilon)D^{-1}L)D^{-\frac{1}{2}}]D^{\frac{1}{2}}(\mathbf{x}^{k,l} - \mathbf{x}^{(k)}_{\text{shift}}) \tag{18}$$

where the middle part can be simplified as

$$\begin{align}
D^{\frac{1}{2}}(I - \frac{\mathbf{1}\mathbf{v}^T}{\mathbf{v}^T\mathbf{1}})(I - (1-\epsilon)D^{-1}L)D^{-\frac{1}{2}} &= (D^{\frac{1}{2}} - \frac{\mathbf{v}\mathbf{v}^T}{\mathbf{v}^T\mathbf{1}})(I - (1-\epsilon)D^{-1}L)D^{-\frac{1}{2}} \tag{19}\\
&= (D^{\frac{1}{2}} - \frac{\mathbf{v}\mathbf{v}^T}{\mathbf{v}^T\mathbf{1}})D^{-\frac{1}{2}}(I - (1-\epsilon)D^{-\frac{1}{2}}LD^{-\frac{1}{2}}) \tag{20}\\
&= (I - \frac{\mathbf{v}\mathbf{1}^T}{\mathbf{v}^T\mathbf{1}})(I - (1-\epsilon)D^{-\frac{1}{2}}LD^{-\frac{1}{2}}) \tag{21}
\end{align}$$

The first equality comes from $D^{\frac{1}{2}}\mathbf{1} = \mathbf{v}$ and the last equality comes from $\mathbf{v}^T D^{-\frac{1}{2}} = \mathbf{1}^T$.

(21) is crucial to our analysis, the first matrix $I - \frac{\mathbf{v}\mathbf{1}^T}{\mathbf{v}^T\mathbf{1}}$ has eigenvalues 1 with multiplicity $n-1$ and 0 with multiplicity 1, the eigenvector corresponding to eigenvalue 0 is $\mathbf{v}$.

The eigenvalues of $D^{-\frac{1}{2}}LD^{-\frac{1}{2}}$ is in range $[0, 2]$. By assuming spectral gap $\rho < 1$, we have $2 - \rho \geq \lambda_1 \geq \lambda_2 \geq \ldots \geq \lambda_{n-1} \geq \rho > \lambda_n = 0$.

Therefore, $I - (1-\epsilon)D^{-\frac{1}{2}}LD^{-\frac{1}{2}}$ has eigenvalues in $\{1, [-1 + 2\epsilon + (1-\epsilon)\rho, 1 - (1-\epsilon)\rho]\}$, where the corresponding eigenvector for eigenvalue 1 is $\mathbf{v}$. However, notice that $(I - \frac{\mathbf{v}\mathbf{1}^T}{\mathbf{v}^T\mathbf{1}})\mathbf{v} = 0$, i.e. this eigenvector will be filtered out by the first matrix.

Therefore, we have

$$\|(I - (1-\epsilon)\frac{\mathbf{v}\mathbf{1}^T}{\mathbf{v}^T\mathbf{1}})(I - D^{-\frac{1}{2}}LD^{-\frac{1}{2}})\| \leq \max\{1 - (1-\epsilon)\rho, 1 - 2\epsilon - (1-\epsilon)\rho\} = 1 - (1-\epsilon)\rho$$

which completes the proof of the proposition.

$$\square$$

# B Proof of Theorem 3.1

**Proposition B.1** *Given two graph Laplacian matrices $L_1, L_2 \in \mathbb{R}^{n \times n}$. Suppose $\lambda_n(L_2) < \lambda_2(L_1)$, then*

$$(L_1 - L_2)^\dagger = L_1^\dagger + \sum_{n=1}^{\infty} (L_1^\dagger L_2)^n L_1^\dagger.$$

*Proof:* By using the identities $L^\dagger L = I - J$, $JL = 0$ in which $L$ is a Laplacian matrix and $J = \frac{1}{n} e \cdot e^T$, we have

$$L_1(I - L_1^\dagger L_2) = L_1 - (I - J)L_2 = L_1 - L_2.$$

Plugging it into $(L_1 - L_2)^\dagger$, we obtain

$$(L_1 - L_2)^\dagger = (L_1(I - L_1^\dagger L_2))^\dagger) = (I - L_1^\dagger L_2)^\dagger L_1^\dagger.$$

With orthogonal decomposition, we can express $L_1, L_2$ as

$$L_1 = \sum_{i=2}^{n} \lambda_i(L_1) u_i u_i^T, L_2 = \sum_{i=2}^{n} \lambda_i(L_2) v_i v_i^T,$$

in which $\{u_i\}, \{v_i\}$ are the corresponding eigenvectors of $L_1, L_2$ respectively. In this way,

$$L_1^\dagger = \sum_{i=2}^{n} \frac{1}{\lambda_i} u_i u_i^T.$$

By definition of quadratic matrix norm, for any vector $x \in \mathbb{R}^n, \|x\|_2 = 1$, we have

$$\| \sum_{i=2} \lambda_i(L_2) v_i v_i^T x \|_2 \leq \lambda_n(L_2),$$

and

$$\| \sum_{i=2} \lambda_i(L_2) v_i v_i^T \sum_{i=2}^{n} \lambda_i(L_1) u_i u_i^T x \|_2 \leq \frac{\lambda_n(L_2)}{\lambda_2(L_1)} < 1,$$

which means $\|L_1^\dagger L_2\|_2 < 1$.

Since $\|L_1^\dagger L_2\|_2 < 1$, the matrix series $M = I + \sum_{n=1}^{\infty} (L_1^\dagger L_2)^n$ converges and $(I - L_1^\dagger L_2)M = I$. Thus $I - L_1^\dagger L_2$ is invertible and its inversion is $I + \sum_{n=1}^{\infty} (L_1^\dagger L_2)^n$. Hence we obtain

$$(L_1 - L_2)^\dagger = L_1^\dagger + \sum_{n=1}^{\infty} (L_1^\dagger L_2)^n L_1^\dagger.$$

$\square$

**Proposition B.2** *Given two $n$ by $n$ Laplacian matrices $L_G$, $L_{G'}$ corresponding to graph $G$, $G'$, which satisfies $G' \subseteq G$. $L_{G'}$ equals to $B_{G'} B_{G'}^T$, where $B_{G'}$ is the edge matrix of $G'$. The maximum degree among all vertices of $G'$ is less or equal to $d$, then we have the following inequality:*

$$\|L_G^\dagger B_{G'}\|_\infty \leq d \left( \max_k |L_{G,kk}^\dagger| + \max_{i \neq j} |L_{G,ij}^\dagger| + \frac{n}{2} \max_{\substack{i,j,k \\ \text{pairwisely different}}} |L_{G,ki}^\dagger - L_{G,kj}^\dagger| \right)$$

*In particular, if $G$ is a clique, the above inequality can be reduced to*

$$\|L_G^\dagger B_{G'}\|_\infty \leq \frac{d}{n}.$$

$$\|L_G^\dagger B_{G'}\|_\infty = \max_k \{\|e_k^T L_G^\dagger B_{G'}\|_1\}$$

$$= \max_k \{\sum_{(i,j)\in G'} |L_{G,ki}^\dagger - L_{G,kj}^\dagger|\}$$

$$\leq \max_k \sum_{\substack{j\in\mathcal{N}(G',k)}} |L_{G,kk}^\dagger - L_{G,kj}^\dagger| + \max_k \sum_{\substack{i,j\neq k \\ (i,j)\in G'}} |L_{G,ki}^\dagger - L_{G,kj}^\dagger|$$

$$\leq d \max_k |L_{G,kk}^\dagger| + \max_k \sum_{j\in\mathcal{N}(G',k)} |L_{G,kj}^\dagger| + \max_k \sum_{\substack{i,j\neq k \\ (i,j)\in G'}} |L_{G,ki}^\dagger - L_{G,kj}^\dagger|$$

$$\leq d \max_k |L_{G,kk}^\dagger| + d \max_{i\neq j} L_{G,ij}^\dagger + \frac{nd}{2} \max_{\substack{i,j,k \\ \text{pairwisely different}}} |L_{G,ki}^\dagger - L_{G,kj}^\dagger|$$

The last line comes from the fact that the size of neighborhoods $\mathcal{N}(G',k)$ is upper bounded by $d$ so the number of edges in $G'$ should be bounded by $\frac{nd}{2}$ as well. $\square$

**Proposition B.3** *Given two $n$ by $n$ Laplacian matrices $L_G$, $L_{G'}$ corresponding to graph $G$, $G'$, which satisfies $G' \subseteq G$. The maximal degree among all vertices of $G'$ is equal or less than $d$, then we claim*

$$\|L_G^\dagger L_{G'}\|_\infty \leq d \left( 2 \max_k |L_{G,kk}^\dagger| + 2 \max_{i\neq j} |L_{G,ij}^\dagger| + n \max_{\substack{i,j,k \\ \text{pairwisely different}}} |L_{G,ki}^\dagger - L_{G,kj}^\dagger| \right)$$

*In particular, if $G$ is a clique, the above inequality can be reduced to*

$$\|L_G^\dagger L_{G'}\|_\infty \leq \frac{2d}{n}.$$

*Proof:*

$$\|L_G^\dagger L_{G'}\|_\infty = \max_k \{\sum_j |\sum_i L_{G,ki}^\dagger L_{G',ij}|\}$$

$$= \max_k \{\sum_j |L_{G,kj}^\dagger deg(G',j) - \sum_{i\in\mathcal{N}(G',j)} L_{G,ki}^\dagger|\}$$

$$= \max_k \{\sum_j |\sum_{i\in\mathcal{N}(G',j)} (L_{G,kj}^\dagger - L_{G,ki}^\dagger)|\}$$

$$\leq 2 \max_k \sum_{(i,j)\in G'} |L_{G,kj}^\dagger - L_{G,ki}^\dagger|$$

$$= 2 \sum_{j\in\mathcal{N}(G',k)} |L_{G,kk}^\dagger - L_{G,kj}^\dagger| + 2 \sum_{\substack{(i,j)\in G' \\ i,j\neq k}} |L_{G,ki}^\dagger - L_{G,kj}^\dagger|$$

$$\leq 2d \max_k |L_{G,kk}^\dagger| + 2d \max_{i\neq j} |L_{G,ij}^\dagger| + n \max_{\substack{i,j,k \\ \text{pairwisely different}}} |L_{G,ki}^\dagger - L_{G,kj}^\dagger|$$

$\square$

*Proof of Theorem 3.1:* At first we show that assuming $\mathbf{x}^{gt} = 0$ will not damage the generality of the proof. To this end, do transformation from original measurements $\mathbf{t}$ to $\mathbf{t'}$ as $t'_{ij} = t_{ij} - x_i^{gt} + x_j^{gt}$. The condition for correct measurement $t_{ij}$ turns to $|t'_{ij}| \leq \sigma$. If $\mathbf{t'}$ with ground truth $\mathbf{0}$, the same truncation strategy as $\mathbf{t}$ with $\mathbf{x}^{gt}$ produces iterative solution $\mathbf{x'}^{(k)}$, and initializaion $\mathbf{x'}^{(0)} = \mathbf{x}^{(0)} - \mathbf{x}^{gt}$, then we assert that

$$\mathbf{x}^{(k)} = \mathbf{x'}^{(k)} + \mathbf{x}^{gt}$$

holds for all $k \in \mathbb{N}$.

By induction we assume $\mathbf{x}^{(k)} = \mathbf{x'}^{(k)} + \mathbf{x}^{gt}$, thus the truncated graph $G_k$ of $\mathbf{x'}^{(k)}$ must be the same as $\mathbf{x}^{(k)}$ since they use the same truncation strategy. Our algorithm provides the next $\mathbf{x}$ and $\mathbf{x'}$ as

$$\mathbf{x}^{(k+1)} = L_{G_k}^\dagger B_{G_k} \mathbf{t}_{G_k}$$

$$\mathbf{x'}^{(k+1)} = L_{G_k}^\dagger B_{G_k} \mathbf{t'}_{G_k}.$$

Since our truncation strategy make sure the correct measurements will not be truncated, $G_k$ is certainly a connected graph. Thus $\mathbf{x}^{gt}$ is the unique precise solution of the linear function $B_{G_k}\mathbf{x} = \mathbf{t}_{G_k}^0$ in which $\mathbf{t}_{G_k}^0$ is the measurements without error on graph $G_k$. Thus by the process of the derivation of $L_{G_k}^\dagger B_{G_k}$ we know that

$$L_{G_k}^\dagger B_{G_k} t_{G_k}^0 = \mathbf{x}^{gt}.$$

Note $\mathbf{t}_{G_k} = \mathbf{t'}_{G_k} + \mathbf{t}_{G_k}^0$, hence we get the identity

$$\mathbf{x}^{(k+1)} = \mathbf{x'}^{(k+1)} + \mathbf{x}^{gt}.$$

We have seen $\mathbf{x}^{(k)}$ with $\mathbf{x}^{gt}$ behaves completely the same as $\mathbf{x'}^{(k)}$ with $\mathbf{0}$, therefore they must have the same concentration bound, so we assume $\mathbf{x}^{gt} = \mathbf{0}$ below.

Returning to the original proposition. Prove this theorem by induction. Assume

$$\|\mathbf{x}^{(k)}\|_\infty \le q\sigma + 2p\epsilon c^{k-1}$$

and

$$k \le -\log\left(\frac{\epsilon(c-4p)}{(1+2q)\,\sigma}\right) / \log c + 1$$

, or

$$(1+2q)\sigma \le \epsilon(c-4p)c^{k-1}.$$

At $k$-th iteration, the truncation threshold should be $c^k\epsilon$. Since for any correct measurement $t_{ij}$ we have

$$|t_{ij} - x_i^{(k)} + x_j^{(k)}| \le \sigma + 2\|\mathbf{x}^{(k)}\|_\infty \le (1+2q)\sigma + 4p\epsilon c^{k-1} \le \epsilon c^k,$$

no correct measurement can be truncated. On the other hand, all survived measurement should satisfy

$$|t_{ij}| \le |t_{ij} - x_i^{(k)} + x_j^{(k)}| + 2|\mathbf{x}^{(k)}|_\infty \le \epsilon c^k + 2q\sigma + 4p\epsilon c^{k-1}$$

Let $G_T$ be the graph consisting of all edges that are dropped at $k$-th iteration. According to the previous argument, we can write $x^k$ as

$$\mathbf{x}^{(k+1)} = (L_G - L_{\overline{G}})^\dagger B_{G\backslash\overline{G}} \mathbf{t}^{(k)}$$

in which $L_G, L_{\overline{G}}$ are the Laplacian matrices of $G$ and $G_T$ respectively while $B_{G\backslash\overline{G}}$ is the edge adjacent matrix of $G\backslash\overline{G}$. It is clear that $G_T \subseteq G\backslash G_a$.

Using Proposition B.1, we have

$$(L_G - L_{\overline{G}})^\dagger = \sum_{k=0}^{\infty} (L_G^\dagger L_{\overline{G}})^n L_G^\dagger.$$

Using Proposition A.2, A.3, we obtain upper bounds

$$\|L_G^\dagger B_{G\backslash G_{good}}\|_\infty \le d_{bad}\alpha = h,$$

$$\|L_G^\dagger L_{\overline{G}}\|_\infty \le 2d_{bad}\alpha = 2h,$$

and

$$\|L_G^\dagger B_{G_{good}}\|_\infty \le n\alpha,$$

so that

$$\|\mathbf{x}^{(k+1)}\|_\infty = \|\sum_{k=0}^\infty (L_G^\dagger L_{\overline{G}})^n (L_G^\dagger B_{G\backslash\overline{G}} \mathbf{t}^{(k)})\|_\infty$$

$$\leq \frac{1}{1-2h}\left\|L_G^\dagger B_{G_{good}} \mathbf{t}_{G_{good}} + L_G^\dagger B_{G\backslash\overline{G}\backslash G_{good}} \mathbf{t}_{G\backslash\overline{G}\backslash G_{good}}\right\|_\infty$$

$$\leq \frac{1}{1-2h}\left(n\alpha\|\mathbf{t}_{G_{good}}\|_\infty + d_{bad}\alpha\|\mathbf{t}_{G\backslash\overline{G}\backslash G_{good}}\|_\infty\right)$$

$$\leq \frac{1}{1-2h}\left(n\alpha\sigma + h(\epsilon c^k + 2q\sigma + 4p\epsilon c^{k-1})\right)$$

$$= q\sigma + p\sigma + p(\epsilon c^k + 2q\sigma + 4p\epsilon c^{k-1})$$

$$\leq q\sigma + 2p\epsilon c^k$$

in which the last line used the inductive condition $(1+2q)\sigma \leq \epsilon(c-4p)c^{k-1}$. The correctness of proposition follows immediately by induction.

Continue the iterative process until $\epsilon(c-4p)c^k < (1+2q)\sigma$ (this means $(1+2q)\sigma \leq \epsilon(c-4p)c^{k-1}$, so our inductive argument works for $x^{(k)}$), at which time we obtain the bound on $x^{(k)}$

$$\|\mathbf{x}^{(k+1)}\|_\infty \leq q\sigma + 2p\epsilon c^k \leq \frac{2p+cq}{c-4p}\sigma$$

$$\square$$

## C  Analysis of the randomized case

### C.1  Proof of Lemma 3.1

The key idea is to leverage the independence of $\{t_{ij}\}$ from $\mathbf{x}^{(0)}$ and redefine the noise model so that the selection of the edges is separated from the measurements. To simplify the notations, we denote

$$\overline{r} = \frac{2\delta}{a+b}, \quad r = \frac{p}{p+(1-p)\overline{r}}.$$

**Lemma C.1** *Associate each edge $(i,j) \in \mathcal{E}$ with a random variable $w_{ij}$ given by*

$$w_{ij} = \begin{cases} 1 & \text{with probability} \quad p + \overline{r}(1-p), \\ 0 & \text{with probability} \quad (1-\overline{r})(1-p) \end{cases} \tag{22}$$

*Redefine the independent measurement $\hat{t}_{ij}$ along each edge as*

$$\hat{t}_{ij} = \overline{t}_{ij} + (1-r)(x_i^{(0)} - x_j^{(0)}), \quad \overline{t}_{ij} := \begin{cases} -(1-r)(x_i^0 - x_j^0) + \sigma U[-1,1] & \text{with probability} \quad r \\ r(x_i^0 - x_j^0) + \overline{r}\zeta_{ij} & \text{with probability} \quad 1-r \end{cases}$$

*Then this noise model and the original model are identical.*

*Proof:* It is clear that the probability that an edge is selected is $p + \frac{2\delta}{a+b}(1-p) = p + (1-p)\overline{r}$. Moreover,

$$w_{ij}\overline{t}_{ij} = \begin{cases} 0 & \text{with probability} \quad p \\ (x_i^{(0)} - x_j^{(0)}) + \overline{r}\zeta_{ij} & \text{with probability} \quad \frac{2\delta}{a+b}(1-p) \\ 0 & \text{with probability} \quad \frac{a+b-2\delta}{a+b}(1-p) \end{cases}$$

$$\square$$

The following proposition provides a decomposition of $\mathbf{x}^{(1)}$ under the new noise model.

**Lemma C.2** *Let $L_w$ and $B_w$ denote the truncated Laplacian matrix and vertex-edge adjacency matrix, respectively. Let $\overline{\boldsymbol{t}}$ collect $\overline{t}_{ij}, (i,j) \in \mathcal{G}$. Then*

$$\boldsymbol{x}^{(1)} = (1-r)\boldsymbol{x}^{(0)} + L_w^+ B_w \overline{\boldsymbol{t}}. \tag{23}$$

*Proof:* Let $\hat{\mathbf{t}}$ collect $\hat{t}_{ij}, (i,j) \in \mathcal{G}$, then

$$
\begin{aligned}
\mathbf{x}^{(1)} = L_{\mathbf{w}}^+ B_{\mathbf{w}} \hat{\mathbf{t}} &= L_{\mathbf{w}}^+ B_{\mathbf{w}}((1-r)B^T \mathbf{x}^{(0)} + \bar{\mathbf{t}}) \\
&= (1-r)(I_n - \frac{1}{n}\mathbf{1}\mathbf{1}^T)\mathbf{x}^{(0)} + L_{\mathbf{w}}^+ B_{\mathbf{w}}\bar{\mathbf{t}} \\
&= (1-r)x^{(0)} + L_{\mathbf{w}}^+ B_{\mathbf{w}}\bar{\mathbf{t}}.
\end{aligned}
$$

$\square$

Since $L_{\mathbf{w}}^+ B_{\mathbf{w}}$ and $\bar{\mathbf{t}}$ are independent, (23) allows us to apply concentration bounds on $\bar{t}_{ij}$. To this end, we first establish the following inequality:

**Lemma C.3** *Denote $\|\boldsymbol{x}^{(0)}\|_{d,\infty} = \max\limits_{1 \le i,j \le n} |x_i^{(0)} - x_j^{(0)}|$ and let $\bar{r}_2 = \max(\|\boldsymbol{x}^{(0)}\|_{d,\infty}, \bar{r})$. For fixed* $\boldsymbol{w}$*, we have*

$$
\mathrm{Var}\big(\boldsymbol{e}_i^T L_{\boldsymbol{w}}^+ B_{\boldsymbol{w}} \bar{\boldsymbol{t}}\big) \le \big((1-r)(\frac{1}{3}+r) \cdot \bar{r}_2^2 + \frac{r\sigma^2}{3}\big) L_{\boldsymbol{w},ii}^+ \tag{24}
$$

*Proof:* First of all, we have

$$
\begin{aligned}
\mathrm{Var}\big(\bar{t}_{ij}\big) &= (1-r)(\frac{\bar{r}^2}{3} + r(x_i^{(0)} - x_j^{(0)})^2) + \frac{r\sigma^2}{3} \\
&\le (1-r)(\frac{\bar{r}^2}{3} + r\|\mathbf{x}^{(0)}\|_{d,\infty}^2) + \frac{r\sigma^2}{3}
\end{aligned}
$$

It follows that

$$
\begin{aligned}
\mathrm{Var}\big(\mathbf{e}_i^T L_{\mathbf{w}}^+ B_{\mathbf{w}} \bar{\mathbf{t}}\big) &= \sum_{(j,k)\in\mathcal{G}} \mathrm{Var}\big(w_{ij}(L_{\mathbf{w},ij}^+ - L_{\mathbf{w},ik}^+)\bar{t}_{ij}\big) \\
&\le \sum_{(j,k)\in\mathcal{G}} w_{ij}(L_{\mathbf{w},ij}^+ - L_{\mathbf{w},ik}^+)^2 (1-r)(\frac{\bar{r}^2}{3} + r\|\mathbf{x}^{(0)}\|_{d,\infty}^2) \\
&= \mathbf{e}_i^T L_{\mathbf{w}}^+ \cdot L_{\mathbf{w}} \cdot L_{\mathbf{w}}^+ \mathbf{e}_i^T (1-r)(\frac{\bar{r}^2}{3} + r\|\mathbf{x}^{(0)}\|_{d,\infty}^2) \\
&= \mathbf{e}_i^T L_{\mathbf{w}}^+ \mathbf{e}_i (1-r)(\frac{\bar{r}^2}{3} + r\|\mathbf{x}^{(0)}\|_{d,\infty}^2) \\
&= (1-r)(\frac{\bar{r}^2}{3} + r\|\mathbf{x}^{(0)}\|_{d,\infty}^2) \cdot L_{\mathbf{w},ii}^+ \\
&\le \big((1-r)(\frac{1}{3}+r) \cdot \bar{r}_2^2 + \frac{r\sigma^2}{3}\big) L_{\mathbf{w},ii}^+.
\end{aligned}
$$

$\square$

Moreover, the range of each summand in $\mathbf{e}_i^T L_{\mathbf{w}}^+ B_{\mathbf{w}} \bar{\mathbf{t}}$ is bounded above by

$$
\max_{1 \le j,k} |L_{\mathbf{w},ij}^+ - L_{\mathbf{w},ik}^+||\max(\bar{t}_{ij}) - \min(\bar{t}_{ij})| \le 2 L_{\mathbf{w},ii}^+ |\max(\bar{t}_{ij}) - \min(\bar{t}_{ij})|
$$

$$
\le 4 L_{\mathbf{w},ii}^+ \bar{r}_2.
$$

The following proposition directly follows from the Bernstein inequality:

**Fact C.1** *Let $\|\mathrm{Diag}(L_{\boldsymbol{w}}^+)\|_\infty = \max\limits_{1 \le i \le n} L_{\boldsymbol{w},ii}^+$. For fixed $\boldsymbol{w}$, we have for*

$$
\Pr\big(\|\boldsymbol{x}^{(1)} - (1-r)\boldsymbol{x}^{(0)}\| \ge \bar{r}_2 \sqrt{\mathrm{Diag}(L_{\boldsymbol{w}}^+)t}\big) \le 2n \exp\Big(-\frac{t^2}{2\big((1-r)(\frac{1}{3}+r) + \frac{4}{3}\sqrt{\mathrm{Diag}(L_{\boldsymbol{w}}^+) \cdot t}\big)}\Big). \tag{25}
$$

It remains to bound the diagonal entries of $L_{\mathbf{w}}^+$, which is given below:

**Lemma C.4** *Let $d_{\min} = \Omega(log^2(n))$ be the minimal degree of $\mathcal{G}$. Suppose $p + \overline{r}(1 - p) = \Omega(\log^2(n)/d_{\min})$. Then w.h.p.,*

$$\|\mathrm{Diag}(L_{\boldsymbol{w}}^+)\|_\infty \leq \frac{1 + o(1)}{(p + \overline{r}(1 - p))d_{\min}\lambda_2(\overline{L}_\mathcal{G})}, \tag{26}$$

*where $\overline{L}_\mathcal{G}$ is the normalized graph Laplacian of $\mathcal{G}$.*

*Proof:* We first show that for the Laplacian matrix $L$ of any graph $G$,

$$L_{ii}^+ \leq \frac{1}{d_i\lambda_2(\overline{L})},$$

where $\overline{L}$ is the normalized graph Laplacian of $G$. In fact,

$$L_{ii}^+ = \frac{1}{d_i}\overline{L}_{ii}^+ \leq \frac{1}{d_i}\lambda_n(\overline{L}^+) = \frac{1}{d_i}\lambda_2(\overline{L}).$$

The rest of the proof follows from the concentration of vertex degrees of random subgraphs and Theorem 1 in [6]. □

Now we can complete the proof of Lemma C.1 by setting $t = O(\sqrt{\log(n)})$ in (25).

## C.2 Proof of Theorem 3.2

As shown in the previous Section, we assume $\mathbf{x}^{gt} = 0$ with losing generality. Lemma 3.1 tells us that for fixed $\mathbf{x}^{(0)}$, one step of TranSync results in a solution that is closer to the ground-truth solution. We can apply it to a dense samples along the segment between $\mathbf{x}^{gt}$ and $\mathbf{x}^{gt} + \frac{a+b}{2}\mathbf{1}$, e.g., $n\sqrt{n}(a + b)/2$ samples so that the distance between adjacent samples along each axis is at most $\frac{1}{n}$. It is clear that Lemma 3.1 still holds among these sample points.

To prove the convergence of $\mathbf{x}^{(k)}$, we seek to bound

$$\|\mathbf{x}^{(k+1)} - \frac{p}{p + (1 - p)c^k}\mathbf{x}^{(k)}\|_\infty \leq \|\overline{\mathbf{x}}^{(k+1)} - \frac{p}{p + (1 - p)c^k}\overline{\mathbf{x}}^{(k)}\|_\infty \tag{27}$$

$$+ \|\mathbf{x}^{(k+1)} - \overline{\mathbf{x}}^{(k+1)}\|_\infty + \frac{p}{p + (1 - p)c^k}\|\mathbf{x}^{(k)} - \overline{\mathbf{x}}^{(k)}\|_\infty, \quad (28)$$

where $\overline{\mathbf{x}}^{(k)}$ is chosen to be the closest point of $\mathbf{x}^{(k)}$ to the segment under the $L^\infty$ norm, and $\overline{\mathbf{x}}^{(k+1)}$ is the result of one step of TranSync. We can apply Lemma 3.1 to obtain a bound on $\|\overline{\mathbf{x}}^{(k+1)} - \frac{p}{p+(1-p)c^k}\overline{\mathbf{x}}^{(k)}\|_\infty$. It remains to bound $\|\mathbf{x}^{(k+1)} - \overline{\mathbf{x}}^{(k+1)}\|_\infty$. To this end, we start with the following Lemma.

**Lemma C.5** *Given a fixed input $\boldsymbol{t}$, starting from two different points $\boldsymbol{x}^{(k)}$ and $\overline{\boldsymbol{x}}^{(k)}$. Let $\boldsymbol{x}^{(k+1)}$ and $\overline{\boldsymbol{x}}^{(k+1)}$ be the results of applying one step of TranSync. Then*

$$\|\boldsymbol{x}^{(k+1)} - \overline{\boldsymbol{x}}^{(k+1)}\|_\infty \leq \frac{2d_{\max}^{dif}\|L_{\boldsymbol{t},\overline{\boldsymbol{x}}^{(k)}}^+\|_{1,\infty}\left(\|\overline{\boldsymbol{x}}^{(k+1)}\|_\infty + \|\boldsymbol{x}^{(k)} - \overline{\boldsymbol{x}}^{(k)}\|_\infty + \frac{1}{2}c^k\right)}{1 - 2d_{\max}^{dif}\|L_{\boldsymbol{t},\overline{\boldsymbol{x}}^{(k)}}^+\|_{1,\infty}}, \tag{29}$$

*where $L_{\mathrm{dif}} = L_{\boldsymbol{t},\overline{\boldsymbol{x}}^{(k)}} - L_{\boldsymbol{t},\boldsymbol{x}^{(k)}}$, and $L_{\boldsymbol{t},\boldsymbol{x}^{(k)}}$ and $L_{\boldsymbol{t},\overline{\boldsymbol{x}}^{(k)}}$) are truncated Laplacians derived from $\boldsymbol{x}^{(k)}$ and $\overline{\boldsymbol{x}}^{(k)}$, respectively. $d_{\max}^{dif}$ is the maximum number of different edges between these two graphs per vertex.*

*Proof:* Let $B_{\mathbf{t},\mathbf{x}^{(k)}}$ and $B_{\mathbf{t},\bar{\mathbf{x}}^{(k)}})$ be the corresponding vertex-edge adjacency matrix. Define $B_{\text{dif}} = B_{\mathbf{t},\bar{\mathbf{x}}^{(k)}} - B_{\mathbf{t},\mathbf{x}^{(k)}}$. First, we have

$$
\begin{aligned}
\|\bar{\mathbf{x}}^{(k+1)} - \mathbf{x}^{(k+1)}\|_\infty &= \|\left(L_{\mathbf{t},\bar{\mathbf{x}}^{(k)}} + L_{dif}\right)^+ (B_{\mathbf{t},\bar{\mathbf{x}}^{(k)}} + B_{dif})\mathbf{t} - L_{\mathbf{t},\bar{\mathbf{x}}^{(k)}}^+ B_{\mathbf{t},\bar{\mathbf{x}}^{(k)}}\mathbf{t}\|_\infty \\
&\leq \|\left(\left(L_{\mathbf{t},\bar{\mathbf{x}}^{(k)}} + L_{dif}\right)^+ - L_{\mathbf{t},\bar{\mathbf{x}}^{(k)}}^+\right) B_{\mathbf{t},\bar{\mathbf{x}}^{(k)}}\mathbf{t}\|_\infty + \|\left(L_{\mathbf{t},\bar{\mathbf{x}}^{(k)}} + L_{dif}\right)^+ B_{dif}\mathbf{t}\|_\infty \\
&= \|\sum_{i=0}^{+\infty} \left(L_{\mathbf{t},\bar{\mathbf{x}}^{(k)}}^+ L_{dif}\right)^i L_{\mathbf{t},\bar{\mathbf{x}}^{(k)}}^+ B_{\mathbf{t},\bar{\mathbf{x}}^{(k)}}\mathbf{t}\|_\infty + \|\sum_{i=0}^{+\infty} \left(L_{\mathbf{t},\bar{\mathbf{x}}^{(k)}}^+ L_{dif}\right)^i L_{\mathbf{t},\bar{\mathbf{x}}^{(k)}}^+ B_{dif}\mathbf{t}\|_\infty \\
&\leq \sum_{i=0}^{+\infty} \|L_{\mathbf{t},\bar{\mathbf{x}}^{(k)}}^+ L_{dif}\|_{1,\infty}^i \left(\|L_{\mathbf{t},\bar{\mathbf{x}}^{(k)}}^+ L_{dif}\|_{1,\infty}^i \|\bar{\mathbf{x}}^{(k+1)}\|_\infty + \|L_{\mathbf{t},\bar{\mathbf{x}}^{(k)}}^+ B_{dif}\mathbf{t}\|_\infty\right) \\
&= \frac{\|L_{\mathbf{t},\bar{\mathbf{x}}^{(k)}}^+ L_{\text{dif}}\|_{1,\infty} \|\bar{\mathbf{x}}^{(k+1)}\|_\infty + \|L_{\mathbf{t},\bar{\mathbf{x}}^{(k)}}^+ B_{\text{dif}}\mathbf{t}\|_\infty}{1 - \|L_{\mathbf{t},\bar{\mathbf{x}}^{(k)}}^+ L_{\text{dif}}\|_{1,\infty}} \\
&\leq \frac{\|L_{\mathbf{t},\bar{\mathbf{x}}^{(k)}}^+\|_{1,\infty} d_{\max}^{dif}\left(2\|\bar{\mathbf{x}}^{(k+1)}\|_\infty + \|\mathbf{t}\|_\infty\right)}{1 - 2d_{\max}^{dif}\|L_{\mathbf{t},\bar{\mathbf{x}}^{(k)}}^+\|_{1,\infty}}
\end{aligned}
$$

Now we can complete the proof as

$$
\|\mathbf{t}\|_\infty \leq \|\mathbf{x}^{(k)}\|_{d,\infty} + c^k \leq 2\|\mathbf{x}^{(k)} - \bar{\mathbf{x}}^{(k)}\|_\infty + c^k.
$$

$\square$

We proceed to control the two remaining quantities $d_{\max}^{dif}$ and $\|L_{\mathbf{t},\bar{\mathbf{x}}^{(k)}}^+\|_{1,\infty}$. In both cases, we leverage the fact that $\bar{\mathbf{x}}^{(k)}$ lies on the line between $\mathbf{0}$ and $\mathbf{1}$ so that we can utilize the independence of $t_{ij}$. We first provide an upper bound on $d_{\max}^{dif}$.

**Lemma C.6** *Suppose $c^k = \Omega(\log^2(n)/n)$. Denote $d_{\max}^{\mathbf{t},\bar{\mathbf{x}}^{(k)}}$ as the maximum degree of graph truncated from $\bar{\mathbf{x}}^{(k)}$, then*

$$
\begin{aligned}
d_{\max}^{dif} &\leq 4\|\boldsymbol{x}^{(k)} - \bar{\boldsymbol{x}}^{(k)}\|_\infty d_{\max}^{\mathbf{t},\bar{\boldsymbol{x}}^k} + \log(n)\sqrt{4\|\boldsymbol{x}^{(k)} - \bar{\boldsymbol{x}}^{(k)}\|_\infty d_{\max}^{\mathbf{t},\bar{\boldsymbol{x}}^k}} \\
&\leq 6\|\boldsymbol{x}^{(k)} - \bar{\boldsymbol{x}}^{(k)}\|_\infty \left(p + (1-p)c^k\right)n
\end{aligned}
\tag{30}
$$

*almost surely.*

*Proof:* Note that the different edges are incurred if each $t_{ij}$ falls in the two intervals $[(x_i^{(k)} - x_j^{(k)}) - c^k, (\bar{x}_i^{(k)} - \bar{x}_j^{(k)}) - c^k]$ and $[(x_i^{(k)} - x_j^{(k)}) + c^k, (\bar{x}_i^{(k)} - \bar{x}_j^{(k)} + c^k]$. The total length of these two intervals is at most $4\|\mathbf{x}^{(k)} - \bar{\mathbf{x}}^{(k)}\|_\infty$. The first inequality directly follows from the Bernstein inequality. The second inequality follows from the fact that maximum vertex degree of a random sub-graph of edge selection probability $p + (1-p)c^k$ is concentrated around $(p + (1-p)c^k)n$. $\square$

The following lemma provides a concentration of $\|L_{\mathbf{t},\bar{\mathbf{x}}^{(k)}}\|_{1,\infty}$.

**Lemma C.7** *Consider random sub-graphs of clique $K_n$ with edge selection probability $q$, we have*

$$
\|L_{\boldsymbol{w}}^+\|_{1,\infty} = \frac{1}{qn}(2 + \frac{O(1)}{\sqrt{q}}).
\tag{31}
$$

*with high probability.*

**Proof:** First of all, for any matrix $A \in \mathbb{R}^{n \times n}$,

$$
\|A\|_{1,\infty} = \max_{1 \leq i \leq n} \sum_{j=1}^n |a_{ij}| \leq \max_{1 \leq i \leq n} \sqrt{n}\sqrt{\sum_{j=1}^n a_{ij}^2} \leq \sqrt{n}\sigma_{\max}(A).
$$

Since the non-zero eigenvalues of $L^+$ fall in-between $[\frac{1}{\lambda_n(L)}, \frac{1}{\lambda_2(L)}]$, it follows that

$$\|L\|_{1,\infty} \leq \|L^+ - \frac{1}{pn}(I_n - \frac{1}{n}\mathbf{1}_n\mathbf{1}_n^T)\|_{1,\infty} + \frac{2}{pn}$$

$$\leq \frac{2}{pn} + \sqrt{n}\max\left(|\frac{1}{\lambda_2(L)} - \frac{1}{pn}|, |\frac{1}{\lambda_n(L)} - \frac{1}{pn}|\right).$$

Since it is well known that the eigenvalues of the graph Laplacian of a Erdős-Rényi graph $G(n,q)$ is concentrated within in the interval $[qn - O(\sqrt{qn}), qn + O(\sqrt{qn})]$, it follows that

$$\|L\|_{1,\infty} \leq \frac{2}{pn} + \frac{O(\sqrt{qn})\sqrt{n}}{p^2 n^2}.$$

$\square$

**Completing the proof of Theorem 3.2.** Now we are ready to prove Theorem 3.2. Denote $\delta_k = \|\mathbf{x}^{(k)} - \bar{\mathbf{x}}^{(k)}\|_\infty$. Combing (29), (30) and (31), we arrive at the following recursion:

$$\delta_{k+1} \leq \frac{15\delta_k(c^k + \delta_k)}{1 - 15\delta_k}. \tag{32}$$

As $\delta_1 = O(\frac{\log(n)}{\sqrt{n}})$. It follows that we can choose a small constant $C_2$ so that

$$\delta_k < p/128, \quad 1 \leq k \leq \min(C_2\sqrt{\log(n)}, \log(\frac{a+b}{2\sigma})/\log(1/(1-p/2))).$$

and

$$\frac{1}{32} \leq c^{C_2\sqrt{\log(n)}} \leq \frac{1}{16}.$$

It then follows from (32) that

$$\delta_k \leq pc^k/4$$

for sufficiently large $n$.

It remains to check $\|\mathbf{x}^{(k+1)}\|_\infty \leq \frac{1}{2}c^{k+1}$. In fact, using (28) we have

$$\|\mathbf{x}^{(k+1)}\|_\infty \leq \frac{(1-p)c^k}{2(p + (1-p)c^k)}c^k + c^k O(\log(n)/\sqrt{n}) + \frac{pc^k}{5} + \frac{p}{p + (1-p)c^k}\frac{pc^{k-1}}{5}$$

$$\leq \frac{1}{2}(1-p/2)c^k$$

$$\leq \frac{1}{2}c^{k+1},$$

which ends the proof of Theorem 3.2.