[Reviews · NeurIPS 2017]

Reviewer 1



Summary: In this paper, the author proposes TranSync algorithm to deal with the 1D translation synchronization problem. In contrast with many other methods, this work considers employing the truncated least squares to alleviate noisy measurements.A theory is presented to analyze the algorithm. Both synthetic and real data are used to demonstrate the efficiency of the proposed approach. Qualitative: The paper is well written and reads easily. The theory is gently exposed while the proofs are left to supplementary materials. The proposed method is demonstrated on a wide set of experiments. The exposition would benefit from a glimpse at the main ideas of the proof in the main text. I like this work.

Reviewer 2



This work deals with translation synchronization that consists in recovering latent coordinate from these noisy pairwise observations. The proposed algorithmic procedure consists in solving a truncated least squares at each iteration, the convergence analysis and exact recovery condition are provided. Finally the experiments illustrate the interest of the proposed strategy compared to state-of-the-art methods. This paper is interesting and complete.

Reviewer 3



In this paper the authors describe a robust and scalable algorithm,TranSync, for solving the 1D translation synchronization problem. The algorithm is quite simple, to solve a truncated least squares at each iteration, and then the computational efficiency is superior to state-of-the-art methods for solving the linear programming formulation. On the other hand, the analyze TranSync under both deterministic and randomized noise models, demonstrating its robustness and stability. In particular, when the pair-wise measurement is biased, TranSync can still achieve sub-constant recovery rate, while the linear programming approach can tolerate no more than 50% of the measurements being biased. The paper is very readable and the proofs of main theorems are clear and appear correct. However, it will be good if the authors can provide more intuition behind the theorem. The numerical experiments are complete and clear.